# Precursor Lesions of Gallbladder Carcinoma: Disease Concept, Pathology, and Genetics

**DOI:** 10.3390/diagnostics12020341

**Published:** 2022-01-28

**Authors:** Yuki Fukumura, Lu Rong, Yifare Maimaitiaili, Toshio Fujisawa, Hiroyuki Isayama, Jun Nakahodo, Masataka Kikuyama, Takashi Yao

**Affiliations:** 1Department of Human Pathology, School of Medicine, Juntendo University, Tokyo 113-8421, Japan; lu.ok@juntendo.ac.jp (L.R.); yifare.li@juntendo.ac.jp (Y.M.); j-nakahodo@juntendo.ac.jp (J.N.); tyao@juntendo.ac.jp (T.Y.); 2Department of Gastroenterology, Graduate School of Medicine, Juntendo University, Tokyo 113-8421, Japan; t-fujisawa@juntendo.ac.jp (T.F.); h-isayama@juntendo.ac.jp (H.I.); 3Department of Gastroenterology, Tokyo Metropolitan Cancer and Infectious Diseases Center, Komagome Hospital, Tokyo 113-0021, Japan; kikuyama110@yahoo.co.jp

**Keywords:** gallbladder carcinoma, pyloric gland adenoma, biliary intraepithelial neoplasia, intracholecystic papillary neoplasm, pancreatobiliary maljunction

## Abstract

Understanding the pathogenesis and carcinogenesis of gallbladder adenocarcinoma is important. The fifth edition of the World Health Organization’s tumor classification of the digestive system indicates three types of preinvasive neoplasm of the gallbladder: pyloric gland adenoma (PGA), biliary intraepithelial neoplasia (BilIN), and intracholecystic papillary neoplasm (ICPN). New terminologies have also been introduced, such as intracholecystic papillary-tubular neoplasm, gastric pyloric, simple mucinous type, and intracholecystic tubular non-mucinous neoplasm (ICTN). Pancreatobiliary maljunction (PBM) poses a markedly high risk for bile duct carcinoma, which was analyzed and investigated mainly by Asian researchers in the past; however, recent studies have clarified a similar significance of biliary carcinogenesis in Western countries as well. In this study, we reviewed and summarized information on three gallbladder neoplastic precursors, PGA, BilIN, and ICPN, and gallbladder lesions in patients with PBM.

## 1. Introduction

Gallbladder cancer is the 22nd most prevalent and 17th most deadly cancer worldwide [1,2,3]. It is noted to disproportionately affect females more than males, perhaps due to the higher propensity of females to have gallstone disease [4]. The incidence of gallbladder cancer is characterized by marked geographic and ethnic variations; there is high to moderate incidence in India, especially northeast and central India, South America (Chile, Bolivia, Colombia), East Asia (Korea, Japan, China), and central Europe (Slovakia, Poland, Czech Republic) [5,6].

Adenocarcinomas of the gallbladder are highly malignant tumors with a dismal prognosis, and clinicopathological and molecular research on their pathogenesis and carcinogenesis is extremely important. According to the American Joint Committee on Cancer, survival of gallbladder adenocarcinoma is based on the staging of the disease, with an 80% five-year survival rate in patients with stage 0 carcinoma in situ lesions and 2% in those with stage 4b disease [7,8], which emphasizes the importance of detection at the early or precursor stage. The advent of laparoscopic cholecystectomy (LC) for gallbladder lesions has been associated with an increased number of incidental gallbladder carcinomas, and some patients with incidental carcinomas may experience trocar metastasis and peritoneal spreading caused by bile spillage during LC, which again emphasizes the importance of establishing preoperative biomarkers for gallbladder carcinoma [9].

As in most epithelial cancers, the development of gallbladder adenocarcinoma is preceded by a variety of precancerous changes. However, compared to other organs, such as colon, uterus, prostate, and pancreas, knowledge of the precursor lesions of gallbladder carcinoma is limited, and their clinicopathologic features are insufficiently characterized. This is partly due to the rarity of these lesions, controversies about the criteria for diagnosis, and the use of several ununified terms when referring to the lesions [10,11,12]. The fifth edition of the World Health Organization’s (WHO) tumor classification of the digestive system (WHO 2019) proposed three types of preinvasive neoplasm (neoplastic precursor) of the gallbladder and unified the terminology: pyloric gland adenoma (PGA), biliary intraepithelial neoplasm (BilIN), and intracholecystic papillary neoplasm (ICPN) [13]. Unifying the terminology in this context seems essential in order for clinicians and researchers to collect and analyze accurate data on these diseases; however, it seems that the present classification and terminology adopted by the WHO 2019 are slightly immature because of the presence of some confusing terms and definitions, especially PGA and ICPN, as described below in this review. Hence, it is important for us to stay up to date on future revisions or additional classifications of gallbladder neoplastic precursors.

We herein review and summarize the present knowledge on the concept, pathology, and molecular features of the three types of neoplastic precursors and gallbladder lesions in pancreatobiliary maljunction of the gallbladder.

## 2. Pyloric Gland Adenoma

PGA is a grossly visible noninvasive neoplasm composed of uniform back-to-back mucinous glands arranged in a tubular configuration [14].

In the fourth edition of WHO’s tumor classification of the digestive system (WHO 2010) and several textbooks, adenomas of the gallbladder have been classified as tubular, papillary (villous), and tubulopapillary (tubulovillous) according to their growth pattern, and cytologically categorized as pyloric gland, intestinal, foveolar, and biliary type [10,11,12]. Hence, tubular adenoma of intestinal type, papillary adenoma of biliary type, and tubular adenoma of pyloric gland type (PGA) were grouped into one “adenoma” category in the WHO 2010 classification [10]. On the contrary, in the WHO 2019 edition, adenomas other than PGA were grouped under the ICPN category, and only PGA was independently classified as one of the three precursor neoplastic lesions of the gallbladder [14]. Details are not given on why, among the grossly visible preneoplastic lesions of the gallbladder, only PGA was placed outside of the ICPN in WHO 2019. However, one possible reason may be that PGA is relatively well-characterized as a neoplastic precursor gallbladder lesion and is more innocuous in behavior compared to other tumors categorized under ICPN [15], hence it is treated as an independent entity.

### 2.1. Clinicopathological Features

Clinicopathological and molecular studies on PGAs are briefly summarized in Table 1. PGAs are found in 0.2–0.5% of gallbladders removed for cholelithiasis or chronic cholecystitis [11], and they account for approximately 10% of all grossly visible noninvasive neoplasms of the gallbladder [16]. PGAs are often <2 cm in size; patients are asymptomatic and the tumors are usually discovered incidentally. There is no specific localization within the gallbladder.

There are no well-established etiological factors associated with PGAs; however, 50–65% are known to be associated with cholelithiasis [17,18]. Occasionally, they occur in association with Peutz–Jeghers syndrome [19] or familial adenomatous polyposis [11].

### 2.2. Gross and Microscopic Pathology

PGAs can occur anywhere in the gallbladder and form sessile or pedunculated polyps (Figure 1A) [17]. Most are solitary; however, cases of multiple PGAs have been reported [11]. Some PGAs contain squamous morules (20%), which are immunohistochemically positive for CDX2 and CD10. Some PGAs contain foamy macrophages, as seen in cholesterol polyps and cholesterolosis of the gallbladder (Figure 1B–D). Immunohistochemically, tumor cells show diffuse positivity for MUC6, while MUC2-positive goblet cells or MUC5AC-positive gastric foveolar metaplastic cells are observable, but are usually scattered (Figure 1E,F) [17,20].

**Table 1 diagnostics-12-00341-t001:** Studies on pathology of pyloric gland adenoma (PGA) of the gallbladder.

	Author ^a^	Patients ^b^	Age ^c^	Tumor Size ^d^	Invasion ^e^	Genetics ^f^	Ref. ^g^
1	Adsay	13	NA	NA	7.6–15.4% *	NA	[16]
2	He	23	62.8 (44–87)	15.1 (5–45)	8.7%	*CTNNB1* (100%) *KRAS* (4.2%) *GNAS* (0%)	[17]
3	Albores-Saavedra	165	NA	NA	0.03%	NA	[18]
4	Wani	29	NA	8.2 (3–16)	NA	NA	[20]
5	Chang	19	NA	NA	0%	*CTNNB1* (80%)	[21]
6	Yanagisawa	17	NA	NA	NA	*CTNNB1* (62.5% **)	[21]

^a^ Name of first author. ^b^ Number of patients with PGA. ^c^ Mean age with range in parentheses. ^d^ Mean tumor size (in millimeters) with range in parentheses. ^e^ Ratio of pyloric gland adenoma with invasive carcinoma. ^f^ Genes with somatic mutations with frequency in parentheses. ^g^ Reference number. * Value is 15.4% when one PGA case with stromal invasion where invasive carcinoma is distant from the polyp is included, 7.6% when it is not included. ** Ratio of CTNNB1 mutation seemed to be investigated among 17 PGAs and 1 intestinal type adenoma.

Recently, a new term, intracholecystic tubular non-mucinous neoplasm (ICTN), was proposed for a type of preinvasive tumor-forming gallbladder neoplasm composed of small, non-mucinous tubules, often with squamoid morules and cholesterolosis in the background, showing diffuse MUC6 expression [22]. According to the authors, PGAs occur due to the proliferation of tumor cells with cytoplasmic mucin, whereas ICTNs are non-mucinous tumor cells. Although the authors mentioned that additional analyses are warranted to investigate ICTNs in order to clarify their cell type, identity, and relationship with other neoplasms occurring in this region, they consider that ICTNs should be regarded separately from ordinary mucinous PGAs. In our opinion, in ordinary PGAs, there is variation in the amount of cytoplasmic mucin, and the appearance of mucin-poor lesions due to focal oncocytic changes and high-grade dysplastic changes are well-known phenomena in PGAs [11]. Considering the common pathologic features of ICTNs and ordinary PGAs, such as occasional squamoid morule formation, diffuse immunohistochemical positivity for MUC6, and frequent *CTNNB1* mutation, ICTN may be a disease entity closely related to PGA.

Neoplastic precursors labeled as PGAs are also present in other organs. Gallbladder PGAs show some histological similarities with PGAs of other organs, especially the stomach, duodenum, and pancreas, where PGA of the pancreas is synonymous with intraductal papillary mucinous neoplasm (IPMN) of the pancreas, gastric pyloric subtype. They are histologically similar in that all of these lesions are mainly composed of mucinous glands resembling gastric pyloric or duodenal Brunner’s glands. Sometimes they contain areas with gastric foveolar and intestinal metaplasia and neuroendocrine cells, showing diffuse expression of MUC6. However, the formation of squamoid morules is limited to gallbladder PGAs, and the underlining molecular abnormalities are quite different between PGAs in the gallbladder and in other areas, which are described below in this review. We need to be careful, because the same term, PGA, is now utilized for different molecular-based neoplasms with some similar histological features [17].

### 2.3. Molecular Features of PGA

Mutations in *CTNNB1* have been reported in 60–100% of PGAs of the gallbladder [17,21,23]. Although they have some histological similarity to PGAs of the stomach, duodenum, and pancreas, the underlying molecular abnormalities are different; gallbladder PGAs frequently have the *CTNNB1* mutation and exceptionally or only rarely have the *KRAS* mutation, and they lack the *GNAS* mutation. On the contrary, PGAs of the stomach, duodenum, and pancreas frequently have *KRAS* and *GNAS* mutations [24,25,26,27].

## 3. Intracholecystic Papillary Neoplasm

ICPN is a recently proposed gallbladder neoplasm characterized as a grossly visible, mass-forming, noninvasive epithelial neoplasm arising in the mucosa and projecting into the gallbladder [28]. Since ICPN was only recently introduced and is relatively rare, knowledge of its clinicopathological features is limited to date, compared to knowledge of IPMN of the pancreas and intraductal papillary neoplasm of the bile duct (IPNB). ICPN is contrasted with BilIN of the gallbladder, in that BilIN is characterized by a microscopically identifiable preinvasive neoplasm and ICPN is a grossly visible mass-forming neoplasm [28].

Although ICPN was originally introduced as an intracholecystic papillary tubular neoplasm, (an umbrella term, according to Pehlivanoglu et al. [23]), in order to address all exophytic, papillary, or polypoid intramucosal masses of the gallbladder, including all adenomas ≥1.0 cm that are composed of preinvasive neoplastic (dysplastic) cells, the WHO 2019 defined ICPN differently; the full term was given as “intracholecystic papillary neoplasm” (without “tubular”). At the same time, PGA was excluded from ICPN and was described as an independent entity, another gallbladder precursor lesion. It is not uncertain why WHO in 2019 utilized the same abbreviation to denote slightly different tumor entities, but we need to be careful which meaning is being referred to when using the term ICPN or reading articles using the term. It should also be noted that in the chapter on PGA in WHO 2019, it is noted is that “intracholecystic papillary-tubular neoplasm, gastric pyloric, simple mucinous type” is related to (or synonymous with) PGA [14], thus introducing another similar term.

There seems to be another issue regarding the entity of ICPN that needs to be resolved in the future: does ICPN include all papillary neoplasms of the gallbladder or only papillary tumors resembling pancreatic IPMNs? Recently, Akita et al. compared ICPNs defined by the more stringent criteria and other papillary carcinomas of the gallbladder; the difference they found was that the former showed more frequent mucin hypersecretion, lacked lymphovascular invasion and nodal metastasis, and were recurrence-free. Using whole-exome and Sanger sequencing, stringently defined ICPN and papillary carcinoma were found to harbor different somatic mutations, as described below [29]. The authors adopted criteria similar to those used for IPNB (IPNB type 1 and 2) [30] and suggested that ICPNs should be restricted to papillary tumors resembling pancreatic IPMNs. On the contrary, Nakanuma et al. referred to all papillary growing tumors as ICPNs, except for BilINs, PGAs, and invasive carcinomas of the gallbladder with considerable intraluminal papillary components, and proposed separating ICPNs into type 1 and 2, as in IPNBs [31].

### 3.1. Clinical Features of ICPN

As mentioned above, the definitions of ICPN used thus far differ depending on the study. Hence, we summarize herein the clinical features of ICPNs reported in each study.

According to the original proposal study by Adsay et al., in 123 cases (including 13 PGAs and 11 non-mucinous pyloric type tumors), the female-to-male ratio was 2:1, which was less striking than the reported female predominance in ordinal pancreatobiliary-type gallbladder adenocarcinomas. The mean patient age for ICPNs was 61 years (range, 20–94 years), which was slightly younger than that for pancreatobiliary-type gallbladder adenocarcinoma (mean age, 64 years) [16]. The average tumor size was 2.6 cm, with a median of 2.2 cm (range, 1.0–7.7 cm). Among the tumors, 70% were solitary and 30% were multifocal.

Akita et al. defined ICPN using stringent criteria and reported a frequency of 6.5% of gallbladder neoplasms that were resected and 35.5% of papillary adenocarcinoma [29]. The authors showed no significant differences between ICPN, papillary carcinoma, and nonpapillary carcinoma in terms of patient age, sex, symptoms, tumor marker level, presence of pancreatobiliary maljunction (PBM), and gallstones; however, several pathologic and genetic features differed, as described below [29].

Nakanuma et al. analyzed 38 cases of ICPN. The authors reported an ICPN frequency of 28% of primary epithelial gallbladder neoplasms and 1.5% of all cholecystectomy cases. They reported stromal invasion frequency of 36.8%, and type 1 and type 2 frequency of 42.1 and 57.9%, respectively [31].

Recently, Mochidome et al. reported that invasive gallbladder adenocarcinoma cases with ICPN components had more favorable prognoses than those without, suggesting that invasive gallbladder carcinoma with exophytic papillary neoplasia may be biologically different from that without such lesions [32]. Two case reports have shown that ICPNs can grow inside Rolitansky–Ashoff sinuses [33] and protrude into the common bile duct [34].

### 3.2. Pathology of ICPN

Pathological studies of ICPN are summarized in Table 2. Histologically, most ICPNs show a mixture of papillary and tubular areas, among which 43% were mostly papillary, 26% were mostly tubular, and 31% were tubulopapillary [16]. ICPNs can be classified into four cell lineages: gastric, intestinal, pancreatobiliary, and oncocytic as in pancreatic IPMN; intra-ampullary papillary-tubular neoplasm (IAPN); and IPNB, although they tend to have more cell lineage diversity than pancreatic IPMNs, and it seems to be more difficult to apply cell lineage-based classification for ICPNs [16]. Histologically, the papillary type tends to be of biliary or mixed type and rarely of gastric or intestinal type. Tubular cases tend to be of gastric lineage or mixed with gastric-type features (Figure 2A–C). Most tubulopapillary cases have a mixed cellular lineage.

Akita et al. showed that grossly detectable mucin hypersecretion was observed significantly more frequently in stringently defined ICPNs than in papillary and non-papillary gallbladder carcinomas. Regarding the tumor cell lineage, all ICPNs were of either gastric or pancreatobiliary type; all intestinal gallbladder carcinoma were papillary or nonpapillary and no gallbladder carcinoma with oncocytic-type morphology was observed in their cohort. Among ICPNs, 43% were noninvasive neoplasms with high-grade dysplasia, and all ICPNs had tumor components with low-grade dysplasia [29]. Recently, a rare ICPN case composed entirely of low-grade dysplasia and without foci of high-grade dysplasia was reported [35].

### 3.3. Molecular Features of ICPN

To date, there has been only one study on the molecular features of ICPN [29]. According to Akita et al., mutations in *STK11* and *CTNNB1* were observed exclusively in ICPNs but not in other papillary or non-papillary carcinomas of the gallbladder, among which the frequency of *STK11* and *CTNNB1* mutations was 43 and 29%, respectively. By contrast, mutations in *TP53* and *ERBB2*/*ERBB3* were more frequent in papillary carcinomas (60% for *TP53*, 80% for *ERBB2*/*ERBB3*) than in ICPNs (14.2% for *TP53*, 14.2% for *ERBB2*/*ERBB3*).

## 4. Biliary Intraepithelial Neoplasia

BilIN is defined as a microscopic, noninvasive, flat or (micro)papillary lesion confined to the bile duct lumen [36]. Clinicopathological studies on BilIN have been conducted mostly with regard to the intrahepatic bile duct, infrequently the distal bile duct, and very rarely the gallbladder. In addition, most knowledge on the molecular features have been obtained from BilINs in the intrahepatic bile duct, and evidence for gallbladder BilIN is limited.

Before 2005, premalignant or noninvasive neoplastic lesions of the intrahepatic and extrahepatic bile ducts were referred to by several terms, such as biliary dysplasia, atypical hyperplasia, and atypical biliary epithelium, and in 2005, a conceptual framework and diagnostic criteria for BilIN were proposed using liver lesions obtained from patients with hepatolithiasis [37]. In 2007, an international interobserver agreement study on the diagnosis of biliary noninvasive neoplastic lesions was conducted to obtain a consensus on the terminology and grading [38].

### 4.1. Pathology of BilIN

Grossly, BilIN lesions are usually visible but may be associated with subtle changes, such as mucosal thickening, granular changes, or rough texture [36,39].

Histologically, BilINs are flat or (micro)papillary epithelial lesions that are graded based on cytoarchitectural atypia. In the WHO 2010, BilIN lesions were classified in a three-tiered system (BilIN-1, -2, and -3); however, a two-tiered system replaced the three-tiered system, with the former BilIN-1 and -2 classified as low-grade and BilIN-3 as high-grade or carcinoma in situ [36]. Whether all gallbladder cancers develop through low-grade and high-grade BilINs is not known with regard to gallbladder carcinogenesis. However, we sometimes observe a morphological transition from low-grade to high-grade BilIN, hence invasive carcinoma, suggesting that at least some gallbladder cancers follow such stepwise carcinogenesis (Figure 3A).

Similar to BilIN lesions in intrahepatic ducts, gallbladder BilINs are relatively diverse histologically; they can be (i) pseudopapillary eosinophilic epithelium with tufting and atypia, (ii) adenoma-like continuously with glandular involvement, or (iii) micropapillary with or without intestinal metaplasia (Figure 3B–D). Although some correlations between BilIN morphology and the preceding diseases/clinical risk factors were suggested for BilINs of intrahepatic bile ducts, this is not the case with gallbladder BilINs.

### 4.2. Molecular Features of Gallbladder BilIN

Regarding gallbladder carcinogenesis, both low-grade and high-grade BilIN lesions have attracted much attention owing to their proposed premalignant behavior, hence stepwise carcinogenesis. There have been several reports of BilINs, including pathological and molecular studies [40,41,42,43,44,45,46]. Morphological studies have suggested the histological transition of low-grade BilIN to gallbladder carcinoma, with or without neighboring high-grade BilIN [40]. In addition, molecular studies have supported stepwise carcinogenesis of the gallbladder through BilIN, including the involvement of *TP53* and *KRAS* mutations in BilIN and dysregulation of p16/cyclin-D1/CDK4 cell cycle pathway in both BilIN and gallbladder cancer [41,42]. Loss of heterozygosity (LOH), defined as a cross-chromosomal event that results in the loss of the entire gene and the surrounding chromosomal region, has been detected in both BilIN and gallbladder cancerous lesions [42,43,44]. LOH in the 3p14.2 locus has been implicated in the pathogenesis of gallbladder carcinogenesis by inactivating the *fragile histidine triad* (*FHIT*) tumor-suppressor gene [45,46]. LOH studies on BilIN are summarized in Table 3.

Interestingly, a recent study conducted by Lin et al. revealed that there are two distinct evolutionary paths for gallbladder cancer: a BilIN-dependent pathway and a BilIN-independent pathway. The authors included 11 cases of T1 gallbladder carcinoma containing both low-grade and high-grade BilIN lesions. By utilizing laser microdissection and separately analyzing each lesion, they inferred the evolutionary relationships among carcinoma and low-grade and high-grade BilIN based on SNV mutations. In the BilIN-independent group, cancerous lesions split before the common ancestor of low-grade and high-grade BilIN. In the BilIN-dependent group, cancerous lesions split after the common ancestor of low-grade and high-grade BilIN, where carcinoma was clustered more closely with high-grade BilIN than low-grade BilIN in some cases and with low-grade BilIN in others [47].

## 5. Pancreatobiliary Maljunction

PBM, also known as abnormal or anomalous junction/union of the pancreatobiliary ductal system, is a congenital malformation with a markedly high risk for biliary cancer. In PBM patients with biliary tract cancers, bile duct and gallbladder cancers develop at a mean age of 50–60 years, which is approximately 15–20 years earlier compared to those without PBM [48,49,50,51]. PBM is defined as a union of the pancreatic and biliary ducts located outside of the duodenal wall, where they merge in the supra-Oddi region. Bile and pancreatic juice reflux and regurgitate mutually. In PBM, because hydropressure is usually higher in the pancreatic duct than in the bile duct, pancreatic juice more frequently refluxes into the biliary duct than bile.

PBM can be divided into two types: with and without congenital biliary dilatation (CBD). The incidence of biliary carcinoma in patients with PBM varies depending on the presence or absence of CBD [48,50,52]. Hence, one of the most controversial problems is what should be the treatment choice for prophylaxis of PBM without CBD. For PBM cases with CBD or with spindle/cylindrical type of dilatation without cancer, excision of the extrahepatic biliary tract with biliary reconstruction is the treatment of choice nowadays. On the contrary, treatment of PBM without an association with biliary dilatation and without cancer is controversial, and there is no fixed viewpoint at present [48,49]. Some surgeons insist that only cholecystectomy should be performed and excision of extrahepatic biliary tract in unnecessary because of the very high incidence of gallbladder cancer and low risk of bile duct carcinogenesis. Others insist that the extrahepatic bile duct must be excised together with the gallbladder from the viewpoint of prophylaxis against carcinogenesis [52]. Regarding the timing of prophylaxis surgery for PBM patients, although no evidence-based recommendations have been established, immediate surgery is recommended when a definitive diagnosis is established, because there are reports of children and young adults with PBM with concurrent cancer [53].

Although PBM has been widely recognized in Asia, it is not in the Western world. Recently Muraki et al. showed that the frequency of PBM in gallbladder cancer patients in the US is approximately 8%, revealing an almost similar occurrence of PBM in gallbladder carcinogenesis as that reported in Japan [54]. The authors pointed out that the PBM cases they detected through their review of radiologic images were not diagnosed as PBM in the original workup.

### 5.1. Clinical Features

According to a nationwide survey in Japan (*n* = 2561), biliary tract cancer is detected in 21.6% of adult PBM patients with CBD and in 42.4% of adult PBM patients without CBD [45,46]. The main tumor location in PBM patients with CBD was the gallbladder in 62.3% of patients and the bile duct in 32.1%, whereas for PBM patients without CBD it was the gallbladder in 88.1% and the bile duct in 7.3%.

### 5.2. Gross and Microscopic Pathology

In PBM patients without cancer, both the gallbladder and bile duct are known to have epithelial hyperplastic changes and frequent metaplasia and dysplasia (Figure 4A,B) [52]. Epithelial papillary hyperplastic changes are diffusely observed just after birth in the gallbladders of patients with PBM (Figure 4C) [54]. Recently, Muraki et al. analyzed 76 gallbladders from patients with PBM, in contrast to patients without PBM, and reported several distinctive histological patterns specific to PBM [55].

The authors compared 76 gallbladder mucosa samples from patients with PBM and 66 from patients without PBM and reported several specific histological findings: (i) diffuse mucosal hyperplasia was significantly more frequent in PBM, (ii) the length of elongated folds was longer (mean: 1.1 cm) in PBM, and (iii) there was the presence of compact villoglandular proliferation, (iv) often broad-based pushing into muscle, (v) more prominent and complex Rokitansky–Ashoff sinus formation, and (vi) frequent horizontal bridging of the folds, bulbous dilatation, and deposition of a peculiar amyloid-like hyaline material at the tips of mucosal hyperplasia. Pyloric gland metaplasia and intestinal metaplasia were also common [56].

Regarding carcinogenesis of the gallbladder in PBM patients, the sequence hyperplasia–dysplasia–carcinoma has been proposed, and this is thought to be different from the adenoma–carcinoma sequence or the de novo carcinogenesis associated with biliary tract cancer in the population without PBM [48]. In the sequence of events of PBM, strong cytotoxic substances (such as lysolecithin) are produced when phospholipase A2 in the pancreatic juice mixes with bile. Consequently, chronic inflammation provokes repeated cycles of damage and healing in the biliary mucosal epithelia. These alterations in the mucosal epithelia (mainly hyperplasia), either alone or in conjunction with DNA mutations, lead to cancer development.

According to recent studies, the histology of gallbladder carcinoma in patients with PBM is more often of the “unusual” type [54,57], where the frequency of sarcomatoid, adenosquamous, and neuroendocrine carcinoma is higher compared to non-PBM gallbladder carcinoma. A recent study by Iwasaki et al. reports a case of ICPN arising in a patient with PBM [58].

### 5.3. Molecular Studies on PBM

Genomic data on gallbladder carcinogenesis have been limited thus far, and data on PBM patients are scarce. Although several whole-exome and next-generation sequencing analyses have been performed on gallbladder carcinoma recently, there are no such analyses on gallbladder carcinoma developing against the background of PBM to date [59,60].

Several studies using target sequences to analyze the role of *KRAS* mutations in PBM and its specificity were conducted, but the conclusions drawn from the results seem controversial. Some previous reports revealed that *KRAS* mutations are detectable not only in cancerous lesions of PBM gallbladder but also in non-cancerous portions in biliary cancer patients with PBM [61,62]. However, other investigators have reported that *KRAS* mutations were detected only in cancerous lesions of PBM gallbladder [63]. In addition, some previous studies reported that *KRAS* mutations are more frequent in PBM-associated events than de novo gallbladder carcinomas [61,62]. In contrast, recent studies showed that the frequency of *KRAS* mutations is <20% in PBM-associated gallbladder cancer, which is comparable to that in non-PBM-associated gallbladder cancer [64]. According to our study data, *KRAS* mutations are not frequent events in gallbladder carcinogenesis in PBM [55]. Regarding the discrepancy in the frequency of *KRAS* mutations, Tomioka et al. suggested that the small numbers of studied cases and differences in analytical methods (most previous studies used the single-strand confirmation polymorphism (SSCP) method, which sometimes causes difficulty in interpretation) may potentially be the reasons [64].

Yamaguchi et al. showed that gallbladder lesions of diffuse papillary hyperplasia (PHP) exhibit senescent features such as expression of p16^INK4A^ and low proliferative activity. This suggests that PHP, which most PBM patients harbor continuously for decades, from infancy through adulthood, without experiencing malignancy, represents a senescence-related lesion against malignant transformation. In this study, the authors also showed that EZH2 was overexpressed in carcinomatous lesions of PBM but not in PHP, suggesting that EZH2 may play a role in escaping from senescence in the gallbladder with PHP [65].

Tomioka et al. showed that interleukin (IL)-33, a member of the IL-1 cytokine family, shows significantly higher expression in PBM-associated cancers than in unassociated cancers of the gallbladder, suggesting that the overexpression or milieu of IL-33 may lead to a pro-oncogenic microenvironment for the gallbladder mucosa in patients with PBM. The pro-oncogenic effects of IL-33 in cholangiocarcinogenesis have been confirmed in recent studies [66], and the authors discussed that blockage of IL-33 may be a less invasive approach to prevent malignant complications in PBM patients [64].

Recent molecular studies on PBM are summarized in Table 4.

## 6. Conclusions

Three precursor neoplastic lesions that were documented in WHO 2019 and gallbladder lesions in PBM patients were summarized in this review. Acknowledging the pathology of precursor neoplastic lesions of the gallbladder is extremely important not only for pathologists but also radiologists, physicians, and surgeons. Although EUS-FNA for diagnosing gallbladder lesions has not been widely used, the demand for the procedure is increasing not only for gallbladder lesions with wall thickening but also for gallbladder polyps [67,68,69].

As a result of the relative rarity, difficulties in obtaining preneoplastic cells and benign background epithelial cells for study, and other reasons, there is little scientific knowledge on gallbladder precursor lesions compared to lesions in other organs [70,71]. We hope recent new technologies will provide precision data regarding precursor neoplastic lesions of the gallbladder in the near future.

## Figures and Tables

**Figure 1 diagnostics-12-00341-f001:**
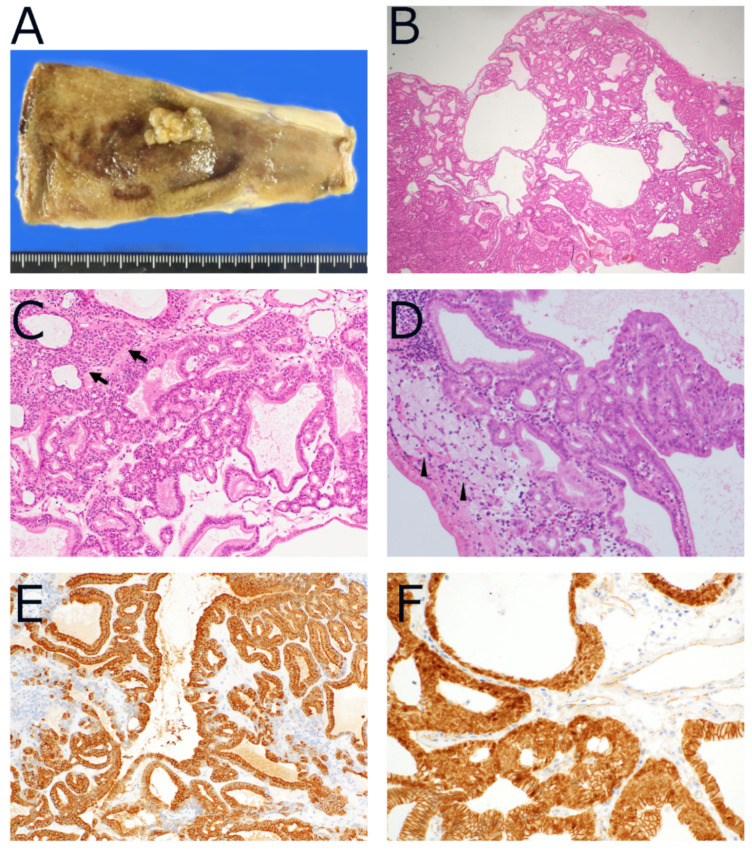
Pathology of pyloric gland adenoma (PGA). (**A**) Sessile-shaped polyp is seen at body of gallbladder. (**B**) Tumor is composed of multiple tubules, some cystically dilated. (**C**) Tumor is composed of cells with intracytoplasmic mucin. Squamoid morules (arrows) are sometimes seen. (**D**) Aggregate of foamy macrophages is seen inside tumor (arrowheads). (**E**) Tumor cells are diffusely positive for MUC6. (**F**) Nuclear expression of β-catenin is often seen. (**B**–**D**) Hematoxylin and eosin, (**E**) immunohistochemistry for MUC6, (**F**) immunohistochemistry for β-catenin.

**Figure 2 diagnostics-12-00341-f002:**
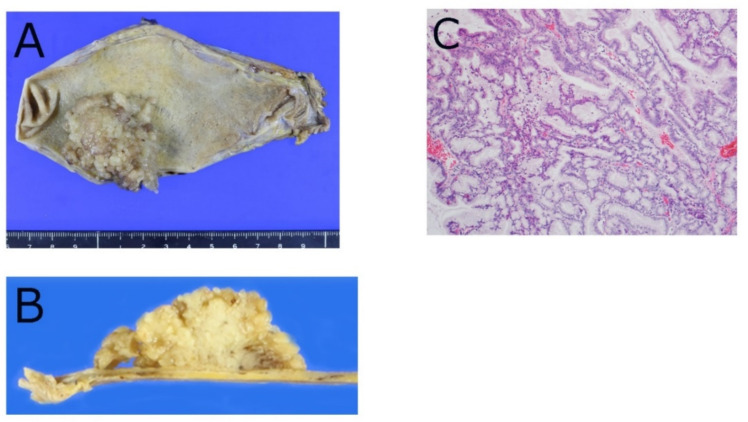
Pathology of intracholecystic papillary neoplasm (ICPN). (**A**) Grossly visible and mass-forming neoplasm arising in gallbladder mucosa in body to fundus of gallbladder. (**B**) Cut section shows exophytic papillary tumor. (**C**) Mostly tubular and sometimes papillary tumor is seen by hematoxyln and eosin staining.

**Figure 3 diagnostics-12-00341-f003:**
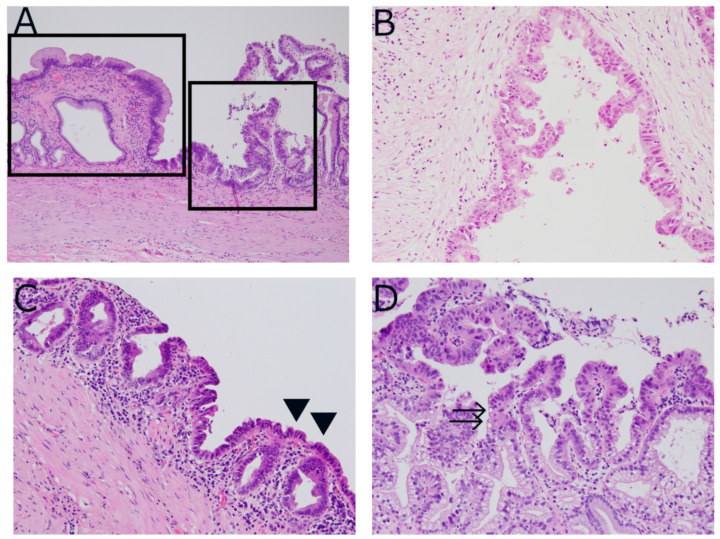
Pathology of biliary intraepithelial neoplasia (BilIN) by hematoxylin and eosin staining. (**A**) Low-grade BilIN component (left box) seen next to high-grade BilIN component (right box). (**B**) High-grade BilIN lesion with pseudopapillary eosinophilic epithelium with tufting and atypia pattern. (**C**) High-grade BilIN with adenoma-like pattern (arrowheads). (**D**) High-grade BilIN with micropapillary pattern. Scattered goblet cells are seen (arrows).

**Figure 4 diagnostics-12-00341-f004:**
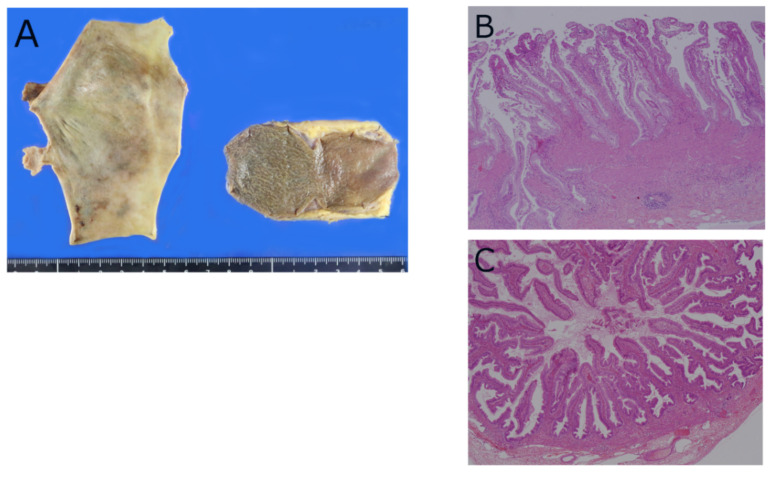
Pathology of gallbladders of patients with pancreatobiliary maljunction (PBM). (**A**) Resected specimens (prophylactic surgery) of gallbladder (right) and dilated bile duct (left). (**B**) Gallbladder mucosa shows diffuse papillary hyperplasia. (**C**) Diffuse papillary hyperplasia observed in gallbladder of 2-year-old girl with PBM. (**B**,**C**) Hematoxylin and Eosin.

**Table 2 diagnostics-12-00341-t002:** Studies on pathology of intracholecystic papillary neoplasm (ICPN).

	Author ^a^	Patients ^b^	Age ^c^	Size ^d^	Invasion ^e^	Genetics ^f^	Ref. ^g^
1	Adsay	123 *	61	2.2	55%	NA	[16]
2	Akita	7 **	72 (61–78)	26 (4–80)	57.1%	*STK11* (43%) *CTNNB1* (29%) *APC* (1%) *KRAS* (1%) *GNAS* (0%) *PIK3CA* (0%)	[29]
3	Nakanuma	38	74.5 ±7.3	28.0 ±14.9	36.8%	NA	[31]

^a^ Name of first author. ^b^ Number of patients with ICPN. ^c^ Mean age with range or standard deviation in parentheses (according to reported style). ^d^ Mean tumor size in millimeters with range or standard deviation in parentheses. ^e^ Proportion of intracholecystic papillary neoplasm with invasive carcinoma. ^f^ Genes with somatic mutations with frequency in parentheses. ^g^ Reference number. * According to originally introduced definition of ICPN, such as ≥1.0 cm and including pyloric gland adenomas. ** Among papillary gallbladder neoplasms, only those with delicate papillary growth and resembling pancreatic intraductal papillary mucinous neoplasms were categorized under ICPN.

**Table 3 diagnostics-12-00341-t003:** Reported loss of heterozygosity (LOH) in biliary intraepithelial neoplasia (BilIN).

	Author ^a^	Patients ^b^	LOH ^c^	Ref. ^d^
1	Kim	5	3p, 5q, 8p, 9p, 13q, 17p, and 18q (80% of BilIN showed LOH at a minimum of one locus)	[42]
2	Wistuba	24	3p (36–86%) 8p (18–58%) 9q (9–58%) 22q (18–53%)	[43]
3	Jain	55	D17S786 *p53* (36.0%) D3S1274 *DUTT1* (25.0%) D3S1766 *FHIT* (23.8%) D5S409 *APC* (54.5%) IFNA *p16* (40.0%) D9S127 *FCMD* (9.1%) D18S34 *DCC* (28.0%) D13S153 *RB1* (4.7%)	[44]
4	Wistuba	26	3p14.2 FHIT (33%)	[46]

^a^ Name of first author. ^b^ Number of patients with BilIN associated with gallbladder carcinoma. ^c^ LOH loci with frequency in parentheses. ^d^ Reference number.

**Table 4 diagnostics-12-00341-t004:** Molecular studies on pancreatobiliary maljunction (PBM)-related gallbladder carcinogenesis.

	Author ^a^	Patients ^b^	Noncancerous Epithelium	Cancerous Epithelium	Ref. ^c^
1	Matsuhara	5	*KRAS*: 33.3%/60% ^θ^ *p53*: 59.3%/60% ^θ^	NA	[61]
2	Nagai	36	*KRAS*: 33.3%/28.6% ^φ^ *p53*: 0%/0% ^φ^ MSI: 0%/85.7% ^φ^	*KRAS*: 60.0% *p53*: 35.3% MSI: 80.0%	[62]
3	Ichikawa	6 *	KRAS:0% Telomerase activity **: 44.4%	KRAS: 66.6% Telomerase activity: 83.3%	[63]
4	Tomioka	32	NA	KRAS: 16% IL-33 mRNA: high	[64]
5	Yamaguchi	15	p16INK4A: high γH2AX: high EZH2: low	EZH2: high	[65]

^a^ Name of first author. ^b^ Number of patients with PBM in total with or without gallbladder carcinoma. ^c^ Reference number. * Several gallbladder foci were analyzed for each case. ^θ^ Values of 33.3 and 59.3% for noncancerous epithelium in PBM cases without cancer and 60 and 60% in PBM cases with cancer. ^φ^ Values of 33.3, 0, and 0% for hyperplastic lesions and 28.6, 0, and 85.7% for dysplastic lesions. ** Telomerase activity was evaluated with fluorescence-based telomeric repeat amplification protocol assay.

## Data Availability

The data supporting the findings of this study are available within the article.

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
