# Peer review of "Precursor Lesions of Gallbladder Carcinoma: Disease Concept, Pathology, and Genetics"

_diagnostics, 2022, doi:10.3390/diagnostics12020341_

Round 1

Reviewer 1 Report

Overall, the manuscript is interesting and helpful for the field, even though not always smooth and clear in the reading. I would suggest minor English editing and the introduction of summary tables.

In the present work, the authors highlighted the importance of unifying the terminology for gallbladder neoplastic precursors; however, the authors discussed that the present classification and terminology adopted by the WHO 2019 is at early stage due to the presence of confusing terminology and definitions. Hence, Fukumura et al., reviewed and summarized the present knowledge on the concept, pathology, and molecular features of the three types of neoplastic precursors in pancreaticobiliary maljunction of the gallbladder.

Overall, the manuscript is interesting and helpful, even though not always smooth and clear in the reading. I would suggest preparing tables summarizing the studies reported in the literature, that would include for example, Authors, Number of patients, Age, Size tumor, Invasive vs non Invasive, Subtype (e.g., Biliary, Gastric, Intestinal, Oncocytic…), Stage, Survival, as well as summary tables with genetic mutations of the three types of neoplastic precursors. When possible, it would be useful to include/provide information about the frequency in men and women.

Minor English editing is required.

Minor comments:

Line 15: Pancreatobiliary. Pancreaticobiliary?

Line 28: Bolvia. Bolivia?

Line 29: …central Europe (Slivakia…Slovakia?)

Line 88: Figure 1A. It is unclear the labeling of Figure 1. I would suggest to revise it and to indicate the magnification for each panel.

Line 94: Figure 1G is missing.

Line 152: Please correct the following “measure ≧1.0 cm”

Line 183: “..the female-to-male ratio was 2.1..”, what does it mean? Please, clarify.

Line 214: Figure 2D is missing.  Figure legend 2 is unclear and lacks information about Figure 2B.

Line 230: Please, also consider:

  1. Muranushi et al., A case report of intracholecystic papillary neoplasm of the gallbladder resembling a submucosal tumor, Surg. Case Report, 2018;
  2. Yokode et al., Intracholecystic papillary neoplasm of the gallbladder protruding into the common bile duct: A case report. Molecular and Clinical Oncology, 2019;
  3. Nakanuma et al., Pathological characterization of intracholecystic papillary neoplasm: A recently proposed preinvasive neoplasm of gallbladder, Annals of Diagnostic Pathology, 2021.

Figure 3: The labeling is barely visible, please adjust.

Figure 3A: I would suggest indicating low vs high grade BilIN component with square boxes and not lines.

Line 278: “There have been several reports of BilINs including pathological and molecular studies”. Please provide references.

References: please check the doubled numbering

Please add the following information to:

Line 424: Author Contributions

Line 425: Funding

Line 426: Institutional Review Board Statement

Line 427: Informed Consent Statement

Line 428: Data Availability Statement

Line 429: Conflicts of Interest

Author Response

Cover letter

To Diagnostics,

To Editor-in-Chief and Editors.

To Mr. Edwin You, Section Managing Editor.

RE: Precursor Lesions of Gallbladder Carcinoma: Disease Concept, Pathology, and Genetics,  IN the special issue “Recent advance in Diagnosis of Gallbladder Diseases.”

I am sending herewith a revised manuscript entitled Precursor Lesions of Gallbladder Carcinoma: Disease Concept, Pathology, and Genetics”. Included are, this cover letter, 1 manuscript (including title page, abstract, text, references, figure legends, 4 tables, and 4 figures.

We authors appreciate for your reviewing our manuscript and giving us lots of thoughtful comments. Below is our point-by-point responses and we highlighted corrected/added contents with yellow in the manuscript. We also highlighted reference # with blue, since we added 3 references.

Reviewer 1

Comment: Overall, the manuscript is interesting and helpful, even though not always smooth and clear in the reading. I would suggest preparing tables summarizing the studies reported in the literature, that would include for example, Authors, Number of patients, Age, Size tumor, Invasive vs non Invasive, Subtype (e.g., Biliary, Gastric, Intestinal, Oncocytic…), Stage, Survival, as well as summary tables with genetic mutations of the three types of neoplastic precursors. When possible, it would be useful to include/provide information about the frequency in men and women.

Our responses: Thank you for your advices. We added 4 tables; Table 1-4.  

Comment: Line 15(⇒16): Pancreatobiliary. Pancreaticobiliary?

Our responses: Thank you for your suggestion. Although some people use pancreatobiliary, while others pancreaticobiliary, we think pancreatobiliary is better for this portion.

Comment: Line 28(⇒29): Bolvia. Bolivia?

Our responses: Thank you for your advice. We corrected it.

Comment: Line 29(⇒29): …central Europe (Slivakia…Slovakia?)

Our responses: Thank you for your advice. We corrected it.

Comment: Line 88: Figure 1A. It is unclear the labeling of Figure 1. I would suggest to revise it and to indicate the magnification for each panel.

Our responses: Thank you for your advice. We revised our Figure 1. 

Comment: Line 94(⇒96): Figure 1G is missing.

Our responses: Thank you for your suggestion. We corrected it. There is no 1G.

Comment: Line 152 (⇒161): Please correct the following “measure ≧1.0 cm”

Our responses: Thank you for your suggestion. We corrected it.

Comments: Line 183(⇒189): “..the female-to-male ratio was 2.1..”, what does it mean? Please, clarify.

Our responses: Thank you for your suggestion. We corrected it.

Comments: Line 214(⇒221): Figure 2D is missing.  Figure legend 2 is unclear and lacks information about Figure 2B.

Our responses: Thank you for your suggestion. We corrected it. There is no 2D. We added information about Figure 2B. 

Comments: Line 230 (⇒209-211): Please, also consider:

  1. Muranushi et al., A case report of intracholecystic papillary neoplasm of the gallbladder resembling a submucosal tumor, Surg. Case Report, 2018;
  2. Yokode et al., Intracholecystic papillary neoplasm of the gallbladder protruding into the common bile duct: A case report. Molecular and Clinical Oncology, 2019;
  3. Nakanuma et al., Pathological characterization of intracholecystic papillary neoplasm: A recently proposed preinvasive neoplasm of gallbladder, Annals of Diagnostic Pathology, 2021.

Our responses: Thank you for your advices. We have read through the above 3 papers, which were very interesting. We referred the 1st and 2nd papers (Muranushi, et al., and Yokode et al., ) as ref. 33 and 34. Also, the 3rd paper (Nakanuma et al.) was already referred as #31.

Comments: Figure 3: The labeling is barely visible, please adjust.

Our responses: Thank you for your suggestion. We revised it.

Comments: Figure 3A: I would suggest indicating low vs high grade BilIN component with square boxes and not lines.

Our responses: Thank you for your suggestion. We revised it.

Comments: Line 278(⇒303): “There have been several reports of BilINs including pathological and molecular studies”. Please provide references.

Our responses: Thank you for your suggestion. We provoded them.

Comments: References: please check the doubled numbering

Our responses: Thank you very much. Checked.

Comments: Please add the following information to:

Line 424 (⇒473): Author Contributions

Line 425: Funding

Line 426: Institutional Review Board Statement

Line 427: Informed Consent Statement

Line 428: Data Availability Statement

Line 429: Conflicts of Interest

Our responses: Thank you very much. We added.

Reviewer 2

Comments: I would like to suggest the Authors to include a brief discussion concerning the role of surgery in occasionally diagnosed galbladder carcinomas in the introduction section (10.1007/s11605-017-3655-z).

Our responses: Thank you very much. We added. (ref. #9).

Comments: I also suggest to add a table to summarize the clinico-pathological and genetic features of the precursor lesions presented in the paper

Our responses: Thank you very much. We made 4 tables.

Sincerely yours,  Yuki Fukumura 2022-1-22.

Corresponding Author and Requests for Reprints:

Yuki Fukumura, M.D., PhD

Department of Human Pathology,

Affiliation: Department of Human Pathology, Juntendo University School of Medicine

Address: Building A 10F, Hongo 2-1-1, Bunkyo-ku, Tokyo, 113-8421, Japan

Telephone: +81-3-5802-1037

Facsimile: +81-3-3812-1056

Reviewer 2 Report

The review paper from Fukumura and coll. entitled "Precursor Lesions of Gallbladder Carcinoma: Disease Concept, Pathology, and Genetics" provides a detailed and up-to-date summary of pre-malignant gallbladder tumors

This well-written and comprehensive review provides interesting informations concerning classification, clinical presentation and etiology of several precursors of gallbladder carcinoma

I would like to suggest the Authors to include a brief discussion concerning the role of surgery in occasionally diagnosed galbladder carcinomas in the introduction section (10.1007/s11605-017-3655-z).

I also suggest to add a table to summarize the clinico-pathological and genetic features of the precursor lesions presented in the paper

Best regards

Author Response

Please see the attachment. Thank you vey much.
